

# An enhanced algorithm for semantic-based feature reduction in spam filtering

María Novo-Lourés, Reyes Pavón, Rosalía Laza, José R. Méndez and David Ruano-Ordás

CINBIO - Biomedical Research Centre, CINBIO, Vigo, Pontevedra, Spain
Galicia Sur Health Research Institute (IIS Galicia Sur), SERGAS-UVIGO, Vigo, Pontevedra, Spain
Department of Computer Science, ESEI - Escola Superior de Enxeñaría Informática, Edificio Politécnico, Universidade de Vigo, Ourense, Ourense, Spain

## ABSTRACT

With the advent and improvement of ontological dictionaries (WordNet, Babelnet), the use of synsets-based text representations is gaining popularity in classification tasks. More recently, ontological dictionaries were used for reducing dimensionality in this kind of representation (*e.g.*, Semantic Dimensionality Reduction System (SDRS) (*Vélez de Mendizabal et al., 2020*)). These approaches are based on the combination of semantically related columns by taking advantage of semantic information extracted from ontological dictionaries. Their main advantage is that they not only eliminate features but can also combine them, minimizing (low-loss) or avoiding (lossless) the loss of information. The most recent (and accurate) techniques included in this group are based on using evolutionary algorithms to find how many features can be grouped to reduce false positive (FP) and false negative (FN) errors obtained. The main limitation of these evolutionary-based schemes is the computational requirements derived from the use of optimization algorithms. The contribution of this study is a new lossless feature reduction scheme exploiting information from ontological dictionaries, which achieves slightly better accuracy (specially in FP errors) than optimization-based approaches but using far fewer computational resources. Instead of using computationally expensive evolutionary algorithms, our proposal determines whether two columns (synsets) can be combined by observing whether the instances included in a dataset (*e.g.*, training dataset) containing these synsets are mostly of the same class. The study includes experiments using three datasets and a detailed comparison with two previous optimization-based approaches.

# INTRODUCTION

Automatic supervised text classification consists of assigning a text document to one (or more) predefined categories. These techniques have been successfully applied to solve a wide variety of problems with specific constraints including spam filtering, language detection, and news categorization (*Kowsari et al., 2019*). The main steps involved in a

Corresponding author
David Ruano-Ordás,
drordas@uvigo.gal

generic strategy for text classification are (*i*) document representation, (*ii*) document pre-processing, (*iii*) feature extraction/selection, (*iv*) model selection, (*v*) training the classifier and (*vi*) executing the classifier on input tests to guess category/categories of each text (*Kowsari et al., 2019*).

Moreover, text classification problems are often divided into three types: (*i*) binary, which involves the use of two mutually exclusive categories (such as spam filtering), (*ii*) multi-class using multiple mutually exclusive categories (language detection), and (*iii*) multi-label where a document can be assigned to zero or several categories from a set of multiple documents (news categorization).

In order to improve text classification accuracy, researchers have studied the use of multiple representation forms (Bag of Words (BoW), character n-grams, word n-grams, *etc.*), text pre-processing techniques (noise removal, spelling correction, stemming, lemmatization, *etc.*), dimensionality reduction schemes (principal component analysis, linear discriminant analysis, evolutionary algorithms, *etc.*) and classification algorithms (Rocchio, Boosting, Bagging, Logistic Regression, *etc.*) (*Kowsari et al., 2019*). However, the introduction of semantic knowledge to aid in the process of automatically classifying texts (*Altınel & Ganiz, 2018*) has significantly contributed to achieving a performance increase in these tasks. Particularly, although the semantic information can be automatically compiled from large collections of documents (topic-based models (*Li et al., 2011*), word embedding (*Wang et al., 2016*)), the use of large ontological dictionaries made by human experts (as WordNet, Babelnet (*Almeida et al., 2016*)) to improve text classification (synset-based representations) has gained popularity (*Sanchez-Pi, Martí & Bicharra Garcia, 2016*; *Lytvyn et al., 2017*; *Ul haq Dar & Dorn, 2018*; *Shanavas et al., 2020*)). Synsets (synonym sets) are groups of words/n-grams with exactly the same meaning which are usually represented by an identifier (*e.g.*, in Babelnet, bn:00008364n ={"*bank*", "*depository financial institution*", "*banking company*", "*banking concern*"}). In synset-based representations, synsets are used as features.

The identification of the right synset when typos are found is a limitation of synset-based representations (*Ruas, Grosky & Aizawa, 2019*). However, the classification models achieve a suitable performance with these kinds of representations. Additionally, researchers have introduced some interesting methods to improve early stages of text classification (representation or dimensionality reduction) by incorporating semantic information from ontologies (information fusion). Many recent works have successfully addressed the reduction of the dimensionality using semantic information achieving (under certain circumstances) an improved performance on classification tasks (*Bahgat et al., 2018*; *Méndez, Cotos-Yañez & Ruano-Ordás, 2019*; *Vélez de Mendizabal et al., 2020*). These schemes are based on grouping features using hypernymy relations extracted from an ontology (Babelnet or WordNet) and are able to link synsets with similar meanings. Often, certain synsets are not included in any text of the training dataset, thus reducing the performance of models. However, by taking advantage of taxonomic relations these approaches bring together these and other semantically similar synsets (which are included in the training dataset) to create a new representation of the dataset, which is semantically richer and, therefore, usually makes classifiers perform better. Additionally,

these approaches keep all knowledge included in the original training datasets (lossless) because their operation is based on combining semantically similar synsets. Hence, the goal of traditional dimensionality reduction methods is to preserve the performance of classifiers while in semantic dimensionality reduction is to improve the accuracy of models.

One of the most important limitations of semantic dimensionality reduction schemes is the time required to achieve a reduction. The work of *Vélez de Mendizabal et al. (2020)* particularly uses MOEA Multi-Objective Evolutionary Algorithms (MOEA) for the identification of feature groups. The authors reported the need for using more than a week to obtain a set of solutions. Therefore, the use of MOEA algorithms to reduce the dimensionality of large datasets is completely unfeasible because the computational requirements increase exponentially. Recently, the same researchers introduced an improvement of the original MOEA algorithm (SDRS low-loss) that was able to slightly increase performance but having the same weaknesses of the original proposal (excessive computational resource consumption) (*Vélez de Mendizabal et al., 2023*).

However, we find that the labelled training dataset contains valuable information which indicates whether two semantically similar synsets (features $s_1$ and $s_2$) can be combined in the same feature $s_{1,2}$ avoiding the need of executing training/testing of a classifier. In particular, features $s_1$ and $s_2$ can be safely grouped if the categories of documents containing $s_1$ and $s_2$ are mainly the same. Based on this idea, we introduce a new efficient semantic-based dimensionality reduction algorithm for binary classification problems (e-SDRS). The algorithm has been implemented and included in our NLPA software (*Novo-Lourés et al., 2020*) and we shared an example of how to use it (*Novo-Lourés, 2022*). We find experimentally that using our dimensionality reduction algorithm, classifiers achieve better accuracy, and their computational requirements are lower. Therefore, our contribution is two-fold: (*i*) the e-SDRS algorithm and (*ii*) an experimental evaluation of its performance and computational resource consumption.

The remainder of the study is structured as follows: 'State of the Art' presents the state of the art in the context of synset-based dimensionality reduction highlighting the limitations of current schemes. 'Enhanced Semantic-based Feature Reduction' introduces our proposal to efficiently address the dimensionality reduction for text classification problems. 'Experimental Evaluation' contains the experimentation made to validate the proposal and finally, 'Conclusions and Future Work' describes the main conclusions and outlines future work.

## STATE OF THE ART

A common problem for text representations to be processed by a classification algorithm is dimensionality. High dimensionality implies an increase in computational costs and resources, as well as a decrease in performance and accuracy due to the inclusion of irrelevant, redundant, and inconsistent information. To solve this problem, different feature selection methods have emerged as alternatives to find the subset of input variables that best describe the collected information (*Kalousis, Prados & Hilario, 2007*). Previous works (*Chandrashekar & Sahin, 2014*; *Deng et al., 2019*), suggest that the classical schemes

used for feature reduction can be classified into three main categories: (*i*) filter methods, (*ii*) wrapper methods, and (*iii*) embedded methods.

Filter methods evaluate the relevance of features present in the input data, usually based on the correlation between the feature and the target variable. Highly ranked features are then selected according to a defined threshold or by setting a maximum number of features. The main advantage of filters is their simplicity; however, they are incapable of selecting independent variables and cannot prevent duplication of information. Examples of filter methods (*Salcedo-Sanz et al., 2004*) include the chi-square test, principal/independent component analysis, mutual information techniques, correlation criteria, and Fisher's discriminant scores.

Wrapper methods use the performance of a classification algorithm to evaluate the amount of relevant information provided by a subset of features. Each subset of features is scored based on its classification performance and the best of them is selected. The main drawback is that it can result in overfitting phenomena.

Embedded methods are those feature selection methods incorporated into specific classifiers and executed during the learning process. The main disadvantage of them lies in their dependency of the learning model.

In contrast to these blind dimensionality reduction mechanisms, recent studies in text classification have shown the utility of taking advantage of semantic information to reduce the dimensionality of input data. The use of semantic information extracted from ontologies (information fusion) for dimensionality reduction is an idea that emerged during the last decade and only a few approaches are available. The first attempt of reducing the dimensionality was introduced in the study of *Bahgat & Moawad (2017)* and it consists of simply using a synset-based representation for the messages. As a synset brings together different terms with the same meaning, this representation implies the realization of a reduction of dimensionality. Although this study is an interesting starting point and helps to connect different texts with similar meanings (I am selling my car, VW Golf for sale, *etc.*), the dimensionality reduction achieved with this technique is very limited. In fact, the process defined by this approach fits the process of translating words to synsets using a word sense disambiguation (WSD) scheme. *Bahgat et al. (2018)* have released a new approach of feature reduction in which two synsets can be merged if they have a parent or child in common and use different weighting schemes based on the ontology relations to decide if more synsets can be merged into the same feature. However, this solution does not adequately evaluate the relevance of the semantic information extracted with and without synsets merging.

The work of *Abiramasundari, Ramaswamy & Sangeetha (2021)* which also focuses on feature reduction, takes advantage of semantic information to decrease the number of words/tokens required for representing messages. Their approach consists of querying a semantic dictionary to check the meaning of each identified word/token. Words/tokens that do not have meaning are removed, resulting in a reduction in the number of features.

In parallel, the study of *Méndez, Cotos-Yañez & Ruano-Ordás (2019)* uses WordNet to reduce the dimensionality of a dataset according to the hierarchical level specified by the user. They select all synsets of WordNet having a distance less than or equal to l from the

synset "*entity*" (the root of the ontological dictionary) as features by using only hypernym relations. Additionally, they provide extensive experimentation using level four of the hierarchy (181 features) achieving quite good classification results. The main limitation of this approach is the lack of using a WSD technique to adequately overcome the problem of polysemy (one word having many distinct meanings). Moreover, this proposal does not consider the class of each text and, therefore, it is not possible to identify those synsets that are relevant for the classification to avoid generalization.

After that, *Vélez de Mendizabal et al. (2020)* used Babelfy to disambiguate the sense of words and raised the reduction of dimensionality as an optimization problem whose main goals were minimising both the number of features and the percentages of FP/FN errors in the classification stage. Then, they took advantage of the Non-dominated Sorting Genetic Algorithm (NSGA-II) (*Deb et al., 2000*) to compute a solution set (Pareto front) and selected the solution achieving the lowest FP/FN errors. Known as SDRS, this approach achieved a significant improvement in the reduction of dimensionality and in the number of errors obtained in the classification process. The main limitation of this approach is the computational requirements of the process. In particular, the evaluation of each candidate solution implied the execution of a 10-fold cross-validation experiment with a naïve Bayes algorithm. This complexity led to the need for weeks of processing to compute interesting solutions.

Additionally, *Saidani, Adi & Allili (2020)* proposed a two-stage semantic analysis-based method to detect domain-specific irrelevant messages. First stage allows grouping the emails according to specific topics to enable a global topic-based conceptual view of spam messages. Following, the second stage then uses rules to identify the semantic meaning of each feature based on the presence of each of the previously specified topics. Authors use six machine learning algorithms to test the performance of the proposed approach.

In early 2023, *Vélez de Mendizabal et al. (2023)* published a new study identifying new dimensionality reduction strategies based on the use of evolutionary computation. This new work introduces a new formulation of the problem in SDRS that can be used for implementing lossless, lossy, and low-loss strategies. The first of them corresponds to the strategy implemented in the original algorithm (SDRS) and allows the evolutionary algorithm group semantically similar synsets with the objectives of reducing classification errors (both FP and FN) and dimensionality. Following the second approach (lossy) the algorithm cannot combine columns, but it is allowed to decide whether to eliminate or keep each column (synset) with the same objectives. Finally, in the low-loss strategy, the evolutionary algorithm has again the same objectives and can achieve them by performing both column combination and deletion operations. Through an experimental analysis, the study concludes that the latter strategy is slightly better than the original SDRS-lossless, but its computational requirements are also excessive.

We carefully studied these algorithms and found that they can improve classification results. The connection of semantic similar synsets helps to slightly generalize knowledge by finding broader features. However, bringing together several synsets should imply that messages containing these synsets are similarly classified. Therefore, supervised dimensionality reduction methods (those that analyse the classifications of a set of

documents) can achieve better results in comparison with other approaches that do not take this information into consideration, such as earlier proposals (*Bahgat & Moawad, 2017*). Moreover, we find that two synsets $(s_1, s_2)$ can be grouped together whenever: (*i*) their semantic distance is short; and (*ii*) most of the documents containing s1 and/or s2 belong to the same class. As long as this condition is very easy to compute, the use of high demanding computational algorithms (such as the above-mentioned SDRS variants) can be avoided.

Finally, a synset-grouping scheme can take advantage of the similarity between two or more synsets $(s_1, s_2, \ldots, s_n)$ to create a synset group even when there is no information about some of the grouped synsets in the training dataset. As an example, and in the context of spam filtering, if the training dataset contains some ham (legitimate) documents with the synset bn:00016606n (cat) and others with bn:00010309n (big cat), the process could include the original synsets (bn:00016606n and bn:00010309n) in the same group. However, the synset bn:00033982n (feline), which is semantically connected with the original symbols (is hypernym for both), should also be included in the group even if there is no training document containing this synset. Reducing the dimensionality with this kind of techniques could successfully minimize the computational requirements of previous approaches and provide enhanced data to improve the classification.

These ideas, which can be easily deduced from observing the operation of previous successful schemes, have been modelled into our proposal (Enhanced Semantic-Based Dimensionality Reduction System, e-SDRS) described in the next section.

## ENHANCED SEMANTIC-BASED FEATURE REDUCTION

This section shows a detailed description of the e-SDRS approach. Our proposal is able to reduce the dimensionality of datasets, which are represented using synset features. It takes advantage of taxonomic relations between synsets (hypernymy and hyponymy) to group similar synsets. Hypernymy/hyponymy relations are included in ontological dictionaries (*e.g.*, Wordnet or Babelnet). Despite any source of information containing hypernymy/hyponymy relations can be used to find these relations, in this work we used Babelnet because it includes Wordnet and Open Multilingual Wordnet (https://omwn.org). Moreover, each synset included in Babelnet has been translated into 520 languages ensuring the same algorithm can be applied with no modifications to datasets written in a large collection of languages.

e-SDRS can be applied only with binary classification. For this purpose, $l_1$ and $l_2$ as exclusive categories for documents. To group synsets, an e-SDRS operation is configured using two parameters: (*i*) *MD* (max distance) and (*ii*) *RS* (required similarity). The former is used to limit the number of taxonomic relations (distance) between two synsets, allowing them to belong to the same group. The latter is used to determine whether the classification for documents that contain two synsets is sufficiently similar (and therefore synsets can be safely grouped or not). Common values for *MD* are included in the interval [2–4] while *RS* values are usually in the range [0.8–0.95].

Let $T$ a set of documents that is being used for training purposes and $docs(S, T)$ a function that computes the collection of documents from $T$ containing any of the synsets

included in the set of synsets $S = \{s_1, s_2, \ldots, s_n\}$. Considering two different synsets $s_1$ and $s_2$ that are present in the set of training documents $(docs(\{s_1\}, T) \neq \varnothing) \wedge (docs(\{s_2\}, T) \neq \varnothing)$, they are indicative of the same label (denoted as iol) when documents in which they are contained mainly belong to the same class $l_i$ (see Eq. (1)). Otherwise, when one or both synsets are not included in $T$, we can consider both synsets to be indicative of the same label.

$$\text{iol}(s_1, s_2) = \left\{ \begin{array}{c} \left( \begin{array}{c} \text{ratio}(docs(\{s_1\}, T), l_1) \geq RS \bigwedge \text{ratio}(docs(\{s_2\}, T), l_1) \geq RS \\ \bigwedge \text{ratio}(docs(\{s_1, s_2\}, T), l_1) \geq RS \end{array} \right) \\ \bigvee \\ \left( \begin{array}{c} \text{ratio}(docs(\{s_1\}, T), l_2) \geq RS \bigwedge \text{ratio}(docs(\{s_2\}, T), l_2) \geq RS \\ \bigwedge \text{ratio}(docs(\{s_1, s_2\}, T), l_2) \geq RS \end{array} \right) \end{array} \right\}. \quad (1)$$

where $\text{ratio}(DS, l)$ stands for the ratio of the documents categorized as $l$ in the document set $DS$.

e-SDRS algorithm is able to create groups of synsets $SG = \{s_1, s_2, \ldots, s_n\}$ that match the following two constraints (Eqs. (C1) and (C2)).

$$docs(\{s_i\}, T) \neq \varnothing \bigwedge \exists s_j \in SG \left| \begin{array}{c} (docs(\{s_j\}, T) \neq \varnothing) \bigwedge (s_i \neq s_j) \\ \bigwedge (\text{dist}(s_i, s_j) \leq MD) \\ \bigwedge \text{iol}(s_i, s_j) \rightarrow s_i \in SG \end{array} \right. \quad (C1)$$

$$docs(\{s_i\}, T) \neq \varnothing \bigwedge \exists s_j, s_k \in SG \left| \begin{array}{c} (docs(\{s_j\}, T) \neq \varnothing) \bigwedge (docs(\{s_k\}, T) \neq \varnothing) \\ \bigwedge \text{iol}(s_j, s_k) \\ \bigwedge (\text{dist}(s_j, s_k) \leq MD), \text{with} s_i \in \text{path}(s_j, s_k) \rightarrow s_i \in SG \end{array} \right. \quad (C2)$$

where $\text{path}(s_1, s_2)$ represents the list of synsets linked by hyponym/hypernym relations which allows the shortest way for connecting $s_1$ and $s_2$ in the ontological dictionary, $\text{dist}(s_1, s_2)$ is the cardinality of the $\text{path}(s_1, s_2)$ reduced in 1.

Intuitively, each group of synsets represents a collection of semantically connected synsets. The training dataset ($T$) contains information for most of them, but the algorithm allows the existence of some synsets in the group that are not included in any document of $T$. However, these will be in the path between two synsets $s_1$ and $s_2$ when (*i*) both are included in one or more documents from $T$, whose distance is lower than $MD$, and (*ii*) the similarity of the labels of documents containing them is greater than $RS$. An example of this situation in the problem of spam filtering is included in Fig. 1.

Figure 1 contains a small subset of synsets included in Babelnet and their hyponym/hypernym relations. For each synset, we represent the information from the training dataset $T$ as percentages (the ratio of documents containing the synset labelled as spam or ham). The training dataset contains documents with synsets *drug* (95% spam and 5% ham), *viagra* (98% spam and 2% ham), *cialis* (91% spam and 9% ham), *medicine* (9% spam and 91% ham), *chloroquine* (92% spam and 8% ham), and *mepacrine* (8% spam and 92% ham).

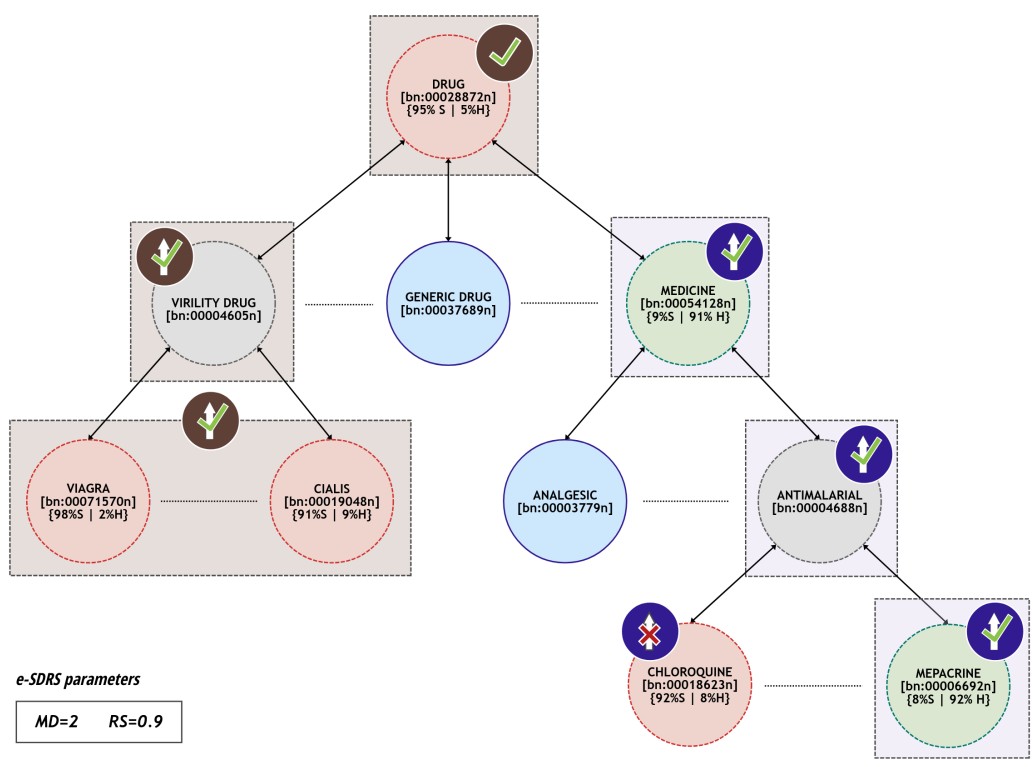

**Figure 1** Example of synsets that can be grouped together.

In the left side of Fig. 1, a group is created with synsets *viagra*, *cialis*, *drug*, and *virility_drug* (represented using brown rectangles). The first three can be successfully grouped with parameters (MD = 2 and RS = 0.9) since the distance between them, considering the taxonomic relations, is two and more than the 90% of the documents containing them are spam (they fit the constraint (C1)). Moreover, *virility_drug* is not included in any document from *T*, but is present in the path that connects at least two of the synsets (in fact, it is in all paths between the other synsets) so it is also included in the group (satisfies constraint (C2)). However, *generic_drug* is not included in the group because it is not included in any path between synsets *viagra*, *cialis* and *drug*. Finally, synsets *chloroquine* and *mepacrine* (included at the bottom right of Fig. 1) cannot be grouped because they are included in documents with considerable dissimilar labelling. However, *mepacrine*, *medicine*, and *antimalarial* (represented using cyan rectangles) could be successfully grouped because they fit the constraints Eqs. (C1) and (C2).

To implement this grouping strategy, e-SDRS comprises two ways of grouping synsets: (*i*) horizontal grouping (denoted as hg) and (*ii*) vertical grouping (represented as vg). Horizontal grouping allows finding synsets that have a common hypernym whilst vertical grouping includes searching synsets that are connected using generalization (hypernymy)

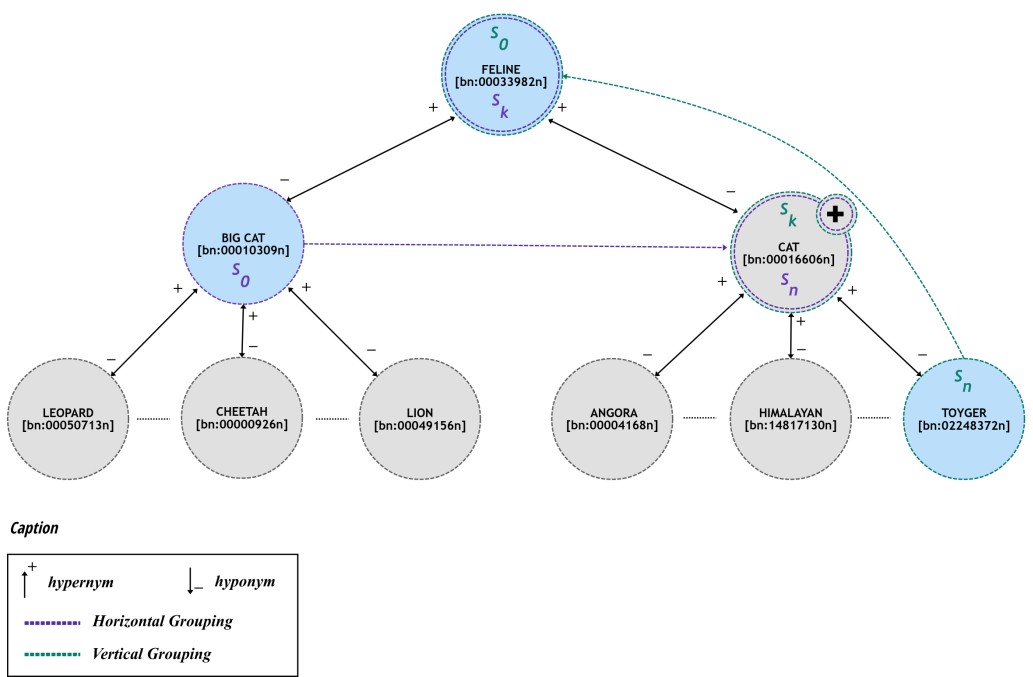

**Caption**

↑+ *hypernym*    ↓− *hyponym*

········ *Horizontal Grouping*

········ *Vertical Grouping*

**Figure 2** **Examples of grouping schemes using in e-SDRS.**

relations. Figure 2 shows two examples of both kinds of grouping forms on the assumption that all documents containing the represented synsets are labelled in the same category.

As shown in Fig. 2, synsets can be grouped by (*i*) using only hypernym relations (vertical grouping) or (*ii*) through the combination of hypernym and hyponym relations (horizontal grouping).

Two synsets $s_0$ and $s_n (n \leq MD)$ included in documents of the training dataset $(\text{docs}(\{s_0\}, T) \neq \varnothing) \wedge (\text{docs}(s_n, T) \neq \varnothing)$ can be vertically grouped ( $\text{vg}(s_0, s_n)$) if there is a path $(s_1, s_2, \ldots, s_{n-1})$, which fulfil the constraint (C3).

$$\exists (s_1, s_2, \ldots, s_{n-1}) \left\lceil \bigwedge_{i=0}^{n-1} \text{hypernym}(s_i, s_{i+1}), \bigwedge_{j,k=0}^{n} \text{iol}(s_j, s_k) \right. \tag{C3}$$

where $\text{hypernym}(s_1, s_2)$ verifies if $s_1$ is a hypernym of $s_2$.

Moreover, a synset $s_0$ included in the training documents $(\text{docs}(\{s_0\}, T) \neq \varnothing)$ and another synset $s_k$ not present in training documents $(\text{docs}(\{s_k\}, T) = \varnothing)$ can be vertically grouped if constraint (C4) is verified.

$$\exists (s_0, s_1, \ldots, s_k, \ldots, s_n) \left| \text{vg}(s_{0,} s_n) \bigwedge_{i=0}^{n-1} \text{hypernym}(s_i, s_{i+1}) \right. \tag{C4}$$

Finally, vertically groupable property (denoted as vg) is transitive and allows creating large groups (see Eq. (2)).

$$\text{vg}(s_0, s_1) \wedge \text{vg}(s_1, s_2) \to \text{vg}(s_0, s_2) \tag{2}$$

Moreover, some synsets can be grouped horizontally (denoted as hg) if they cannot be grouped vertically as shown in constraint (C5).

$$vg(s_0, s_n) \rightarrow \neg hg(s_0, s_n). \tag{C5}$$

Some situations can be considered when grouping synsets horizontally. The first happens when synsets are included in the training dataset $(docs(\{s_0\}, T) \neq \varnothing) \wedge (docs(\{s_n\}, T) \neq \varnothing)$ with $(n \leq MD)$. Two synsets in this situation can be included in the same group $(hg(s_0, s_n))$ if they match the constraint (C6).

$$\exists (s_1, s_2, \ldots, s_{n-1}) \left| \bigwedge_{i=0}^{n-1} \left( hypernym(s_i, s_{i+1}) \vee hyponym(s_i, s_{i+1}) \right), \bigwedge_{j,k=0}^{n} iol(s_j, s_k). \tag{C6}$$

Where $hypernym(s_1, s_2)$ and $hyponym(s_1, s_2)$ are true when $s_1$ is hypernym or hyponym of $s_2$, respectively.

Additionally, a synset $s_0$ included in the training dataset $(docs(\{s_0\}, T) \neq \varnothing)$ can be horizontally grouped with another synset $s_n$ that is not included in $T$ $(docs(\{s_n\}, T) = \varnothing)$ but is included in a vertical group $\left( \exists s_m | (docs(s_m, T) \neq \varnothing) \wedge vg(s_m, s_n) \right)$. Constraint (C7) is required for grouping synsets in such situations $\left( hg(s_0, s_n) \right)$ where the labelling of synset $s_n$ is guessed by one of the members of the vertical group $(s_m)$.

$$\exists (s_1, s_2, \ldots, s_{n-1}) \left| \begin{array}{l} \bigwedge_{i=0}^{n-1} \left( hypernym(s_i, s_{i+1}) \vee hyponym(s_i, s_{i+1}) \right) \\ \bigwedge_{j,k=0}^{n} iol(s_j, s_k) \\ \bigwedge iol(s_0, s_m) \end{array} \right. . \tag{C7}$$

A similar situation occurs when none of the synsets $s_0, s_n$ are included in the training dataset $(docs(\{s_0\}, T) = \varnothing) \wedge (docs(\{s_n, T\}) = \varnothing)$ but are vertically grouped $\exists s_m, s_l | vg(s_m, s_0) \wedge vg(s_l, s_n) \wedge (docs(\{s_m\}, T) \neq \varnothing) \wedge (docs(\{s_l\}, T) \neq \varnothing)$. As shown in constraint (C8), the labelling of these synsets is guessed by the members of their vertical groups $(s_m, s_l)$ to check if these synsets can be horizontally grouped $\left( hg(s_0, s_n) \right)$.

$$\exists (s_1, s_2, \ldots, s_{n-1}) \left| \begin{array}{l} \bigwedge_{i=0}^{n-1} \left( hypernym(s_i, s_{i+1}) \right) \vee hyponym(s_i, s_{i+1}) \\ \bigwedge_{j,k=0}^{n-1} iol(s_j, s_k) \\ \bigwedge_{v=1}^{n-1} docs(\{s_v\}, T) \neq \varnothing \rightarrow (iol(s_v, s_l) \wedge iol(s_v, s_m)) \end{array} \right. . \tag{C8}$$

Considering a synset $s_k$ not included in the training dataset $(docs(\{s_k\}, T) = \varnothing)$ and not included in vertical relations, and another synset $s_0$ $(docs(\{s_0\}, T) \neq \varnothing)$, both can be horizontally grouped if $s_k$ is in the path between $s_0$ and $s_n$, and these synsets are horizontally groupable (constraint (C9)).

$$\exists (s_0, s_1, \ldots, s_k, \ldots, s_n) \left| hg(s_0, s_n) \bigwedge_{i=0}^{n-1} \left( hypernym(s_i, s_{i+1}) \vee hyponym(s_i, s_{i+1}) \right). \tag{C9}$$

Finally, hg (horizontally groupable property) satisfies transitive property, which guarantees the condition expressed in (3) and ensures the generation of large groups.

$$hg(s_0, s_1) \wedge hg(s_1, s_2) \rightarrow hg(s_0, s_2). \tag{3}$$

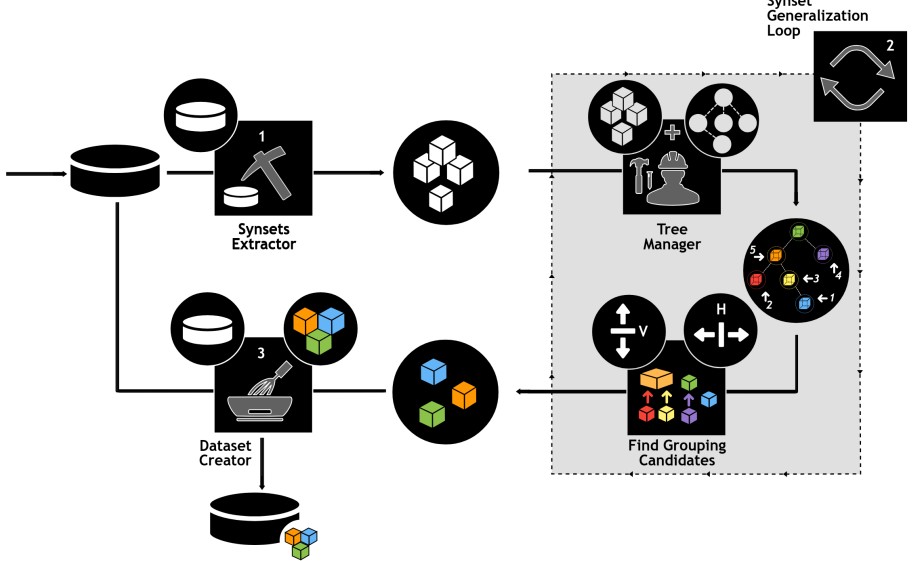

**Figure 3 Using e-SDRS algorithm.** Icons from Noun Project (CCBY3.0): https://thenounproject.com/icons/.

The use of e-SDRS algorithm (which uses the concept of vertical and horizontal grouping) is represented in Fig. 3.

As shown in Fig. 3, before using the e-SDRS algorithm, the original dataset should be represented using synset-based properties (stage 1). During stage 2, each of the synsets (columns) identified are stored in a tree structure according to Babelnet hypernym/hyponym relations (see Tree Manager on Fig. 3). The identification of groups starts from leaf nodes having the largest distance from root to build larger synset groups and avoid removing relations caused by grouping nodes at higher layers. In this manner, each synset is compared with the remaining ones to identify whether vertical grouping can be applied. Then, a similar process is carried out to discover horizontal grouping opportunities. The process is repeated until no grouping is found. Once dimensionality of the training dataset has been reduced, each test instance (or an entire test dataset) can be minimized following the scheme found for the training dataset (Dataset Creator in Fig. 3). Finally, we would point out that, depending on the information included for each column for documents (frequency/presence or absence, *etc.*), the grouping of columns should be implemented in a different form (*i.e.,* using a numerical sum or logical OR operator).

The algorithm has been implemented and included in our NLPA software (*Novo-Lourés et al., 2020*). Using this software, we designed an experimental evaluation protocol to test the suitability of e-SDRS. 'Experimental Evaluation' includes the details of our experimentation.

**Table 1  Available datasets.**

| Type | Dataset name | Number of entries | Spam ratio | References |
|---|---|---|---|---|
| | Ling-Spam | 2.893 | 17% | *Sakkis et al. (2003)* |
| | Enron Spam | 36.715 | 45% | *Metsis, Androutsopoulos & Paliouras (2006)* |
| | PU1 | 1.099 | 44% | |
| | PU2 | 721 | 20% | *Androutsopoulos et al. (2000)* |
| | PU3 | 4.139 | 44% | |
| | PUA datasets | 1.142 | 50% | |
| | Spambase Dataset | 4.601 | 39% | *Newman & Merz (1998)* |
| E-mail | SpamAssassin | 6.047 | 31% | *Apache SpamAssassin Project (2005)* and *Pérez-Díaz et al. (2012)* |
| | CSDMC2010 SPAM corpus | 4.307 | 32% | *CSMINING Group (2010)* |
| | TREC 2005 | 92.189 | 57% | |
| | TREC 2006 | 37.822 | 66% | *NIST (2007)* |
| | TREC 2007 spam corpus | 75.419 | 67% | |
| | SMS Spam collection | 5.574 | 13% | *Almeida, Hidalgo & Yamakami (2011)* |
| SMS | British English SMS corpora | 875 | 49% | *Nuruzzaman, Lee & Choi (2011)* |
| Social | Youtube Spam Collection | 1.956 | 51% | *Almeida et al. (2016)* |
| | TAMU Social Honeypot Dataset | 22.223 | 42% | *Lee, Eoff & Carvelee (2011)* |
| | UK-2006 | 7.473 | 20% | |
| | UK-2007 | 6.479 | 6% | *Castillo (2007)* and *Wahsheh et al. (2012)* |
| | UK-2011 Web Spam Dataset | 3.700 | 23% | |
| Web | ClueWeb09 | 1.040.809.705 | 18% | *Croft & Callan (2016)* |
| | ClueWeb12 | 2.820.500 | 11% | |
| | Webb spam 2011 | 350.000 | 100% | *Wang, Irani & Pu (2012)* |
| | ECML/PKDD 2010 DiCDS | 191.388 | 8% | *Benczúr et al. (2010)* |

## EXPERIMENTAL EVALUATION

In this section we analyse the performance achieved by using our proposal (e-SDRS) and compare it with those achieved by using lossless and low-loss SDRS optimization-based variants. We discarded the use of older simple approaches (*e.g.*, the studies by *Bahgat et al., 2018* and *Méndez, Cotos-Yañez & Ruano-Ordás, 2019*) because they present a lot of limitations that have already been discussed in 'State of the Art'. The original e-SDRS implementation has been designed for its use in binary classification problems. One of the better-known problems in this domain is the filtering of spam messages, which we selected for the experiment. Table 1 compiles a list of available corpora in the problem of spam filtering.

We have used the YouTube Spam Collection and SpamAssassin corpus datasets to experimentally determine the best configuration values for the MD and RS parameters. Combining the results obtained with a small and a large dataset may be suitable for finding the appropriate values for these parameters. Then, the Ling-Spam dataset was used to test

the performance of e-SDRS using the previously determined configuration parameters. This is a medium-sized dataset widely used for testing spam filtering techniques. Finally, we have also used the YouTube Spam Collection dataset to compare the performance obtained in this study and previous (computationally intensive) approaches.

To carry out the experimentation, the YouTube Spam Collection, SpamAssassin corpus and Ling-Spam datasets were represented using synsets. Following this representation, the dimensionalities achieved for these datasets are 1656 (using traditional token-based representations, 1950 tokens/words), 39,435 (57,117 tokens/words) and 39,174 (60962 tokens/words) synset features, respectively. The preprocessing was executed using NLPA software (*Novo-Lourés et al., 2020*) and include the following steps: (i) strip HTML tags, (ii) expand contractions, (iii) expand acronyms, (iv) translate slang terms, (v) remove interjections, (vi) remove emojis, (vii) remove emoticons, (viii) remove stopwords (and words having 3 or less characters), (ix) transform text to lowercase, (x) transform text to synset sequences and (xi) build feature vectors using frequency (number of occurrences of the synset divided by total number of synsets found in document).

To adequately test the performance achieved by the e-SDRS algorithm, we designed the experimental protocol shown in Fig. 4.

As shown in Fig. 4, the experimental protocol measures the theoretical performance that can be achieved and the performance in a production environment. The theoretical evaluation was included to determine the maximum achievable performance of each feature selection scheme. This efficiency could only be achieved when all instances (including those reserved for testing purposes) are available during feature selection. Therefore, in this ideal scenario, we used 100% of the instances for feature selection and after that, the dataset is split into two parts (75% and 25%) which were used training and testing purposes, respectively. Because this ideal situation could not be achieved in practice (know all instances during feature selection), we simulated a real scenario by dividing the dataset before the feature selection stage into the same parts (75% and 25%). Then, 75% of the instances were used for both dimensionality reduction and training purposes. The remaining instances (25%) were transformed according to the results of the feature selection scheme and used for testing purposes. Using instance identifiers, we kept the same distribution of instances when splitting the dataset (75%/25%) in theoretical and real scenarios. To ensure comparability of results, the outcomes in the theoretical and real scenarios were compared with those from a baseline scenario, where no dimensionality reduction scheme was used.

The experiment included the use of several classification schemes including nb (a naïve Bayes classifier provided in klaR R package (*Weihs et al., 2005*)), ranger (a fast implementation of random forests particularly suited for high dimensional data (*Wright & Ziegler, 2017*)), rpart (Recursive partitioning for classification[‡]), Support Vector Machines with polynomial kernels (implemented in kernlab R package), and treebag (Bagging of Classification and Trees) (*Sutton, 2005*). To facilitate the experimentation, we took advantage of the functions provided by caret R package (*Kuhn, 2008*). Additionally, the parameters used for classifiers were optimized by using caret functionalities. We discarded the usage Deep Learning-based schemes in the experimentation because they require the use of word-embedding representations which are incompatible with synset-based ones.

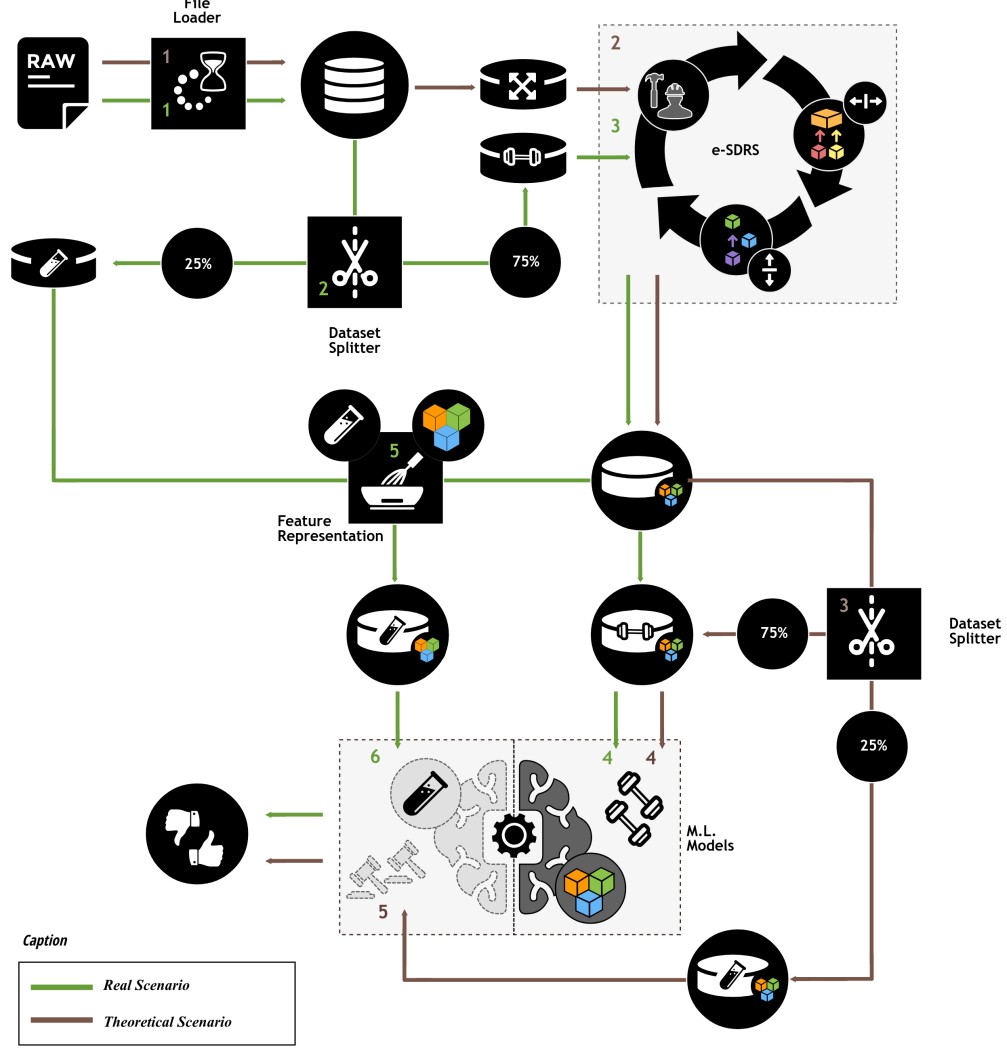

**Figure 4** **Experimental protocol.** Icons from Noun Project (CCBY3.0): https://thenounproject.com/icons/.

Different values of RS were used for the experiment, including 0.95, 0.90, 0.85, and 0.80. The MD parameter was set to values 2, 3 and 4. Our first experiment was aimed at evaluating the effectiveness of the different configurations (RS and MD) of the e-SDRS algorithm using Kappa coefficient (*Cohen, 1968*) to select the most suitable configuration for these parameters. For comparison purposes, the results obtained with e-SDRS were compared with those obtained without using any dimensionality reduction scheme (baseline). Table 2 presents the results achieved using the theoretical scenario with YouTube Comments and SpamAssassin datasets. We also measured the percentage of features reduced in each configuration and represented them inside square brackets.

As shown, the classification accuracy improved when using most e-SDRS configurations. However, although the theoretical scenario allows seeing the maximum capacity of the

**Table 2** Evaluation of e-SDRS performance in a theoretical scenario.

**(a) Youtube Comments Dataset.**

| | MD2 | | | | MD3 | | | | MD4 | | | |
|---|---|---|---|---|---|---|---|---|---|---|---|---|
| | RS80 [5.92%] | RS85 [5.92%] | RS90 [5.8%] | RS95 [5.68%] | RS80 [12.6%] | RS85 [12.68%] | RS90 [12.44%] | RS95 [12.38%] | RS80 [14.01%] | RS85 [13.95%] | RS90 [13.04%] | RS95 [13.04%] |
| nb | +0.0506 | +0.0506 | +0.0506 | +0.0255 | +0.2129 | +0.1825 | +0.1941 | +0.1825 | +0.2055 | +0.1825 | +0.1593 | +0.1593 |
| ranger | +0.0681 | +0.0590 | +0.0787 | +0.0376 | +0.1862 | +0.1714 | +0.1714 | +0.1623 | +0.1813 | +0.1913 | +0.1822 | +0.1913 |
| rpart | +0.0808 | +0.0471 | +0.0778 | +0.0128 | +0.0432 | +0.2089 | +0.1778 | +0.2089 | −0.0222 | −0.0340 | +0.0918 | +0.0696 |
| svmPoly | −0.5036 | −0.5036 | +0.0938 | +0.0618 | +0.2503 | −0.0578 | +0.1297 | +0.2203 | +0.2499 | +0.2396 | +0.2421 | −0.5674 |
| treebag | +0.0521 | +0.0874 | +0.0712 | +0.0641 | +0.2088 | +0.1806 | +0.2177 | +0.1874 | +0.2016 | +0.2105 | +0.2006 | +0.2006 |

**(b) SpamAssassin Dataset**

| | MD2 | | | | MD3 | | | | MD4 | | | |
|---|---|---|---|---|---|---|---|---|---|---|---|---|
| | RS80 [7.38%] | RS85 [5.45%] | RS90 [4.42%] | RS95 [3.21%] | RS80 [7.82%] | RS85 [5.85%] | RS90 [4.7%] | RS95 [3.44%] | RS80 [8.08%] | RS85 [6.11%] | RS90 [4.8%] | RS95 [3.49%] |
| nb | +0.0686 | +0.0541 | +0.0415 | +0.0229 | +0.1588 | +0.0563 | +0.0499 | +0.0394 | +0.0788 | +0.0644 | +0.0537 | +0.0394 |
| ranger | +0.0158 | −0.0038 | −0.0127 | +0.0209 | +0.0317 | −0.0100 | −0.0251 | +0.0164 | +0.0096 | +0.0237 | −0.0021 | +0.0238 |
| rpart | +0.0564 | +0.0246 | +0.0440 | +0.0411 | +0.0706 | +0.0372 | +0.0485 | +0.0460 | +0.0625 | +0.0444 | +0.0451 | +0.0407 |
| svmPoly | −0.3724 | +0.1374 | +0.1234 | +0.1307 | +0.1898 | −0.7733 | −0.7733 | −0.0429 | +0.1526 | +0.1712 | +0.0103 | −0.7733 |
| treebag | +0.0182 | +0.0062 | −0.0029 | +0.0142 | +0.0349 | +0.0051 | +0.0003 | +0.0148 | +0.0196 | +0.0156 | +0.0095 | +0.0101 |

technique, it does not fully represent a real content filtering environment where the messages to be classified are unknown at the time of deciding the attributes (synsets) to be used for message representation. The comparison of e-SDRS configurations in a real scenario is shown in Table 3.

Summarizing information from Table 2, 103 out of 120 total e-SDRS configurations in the theoretical scenario (five algorithms, three different values of MD, four different values of RS and two datasets) were useful to improve the classifier performance. Furthermore, as shown in Table 3, the ratio of configurations improving or maintaining the performance was reduced to 71/120. The best parameters found in the above experiments was MD = 3 RS = 80, which were useful to improve the results in 10 of 10 analysed classifiers and datasets in the theoretical scenario and 8 of 10 in the real scenario. Using this configuration was useful to run the additional performance analyses presented below.

After this, we measured the ratios of correctly classified messages (OK), FP and FN errors. The measurements were made using the best configuration detected in the previous step for theoretical and real scenarios. These scenarios were compared with another one in which the e-SDRS feature reduction scheme was not used (baseline). Results for both selected datasets were plotted in Fig. 5.

As shown, the error ratios (especially FP ones) slightly decrease when using e-SDRS in almost all situations due to the successful incorporation of semantic information from Babelnet. As an exception, the SVM classifier evaluated in the real scenario with YouTube Comments dataset achieves poor performance. Although SVM classifiers can achieve high accuracy levels, their performance could decrease when the data used for training is too sparse (*Yang et al., 2017*). Due to the small length of YouTube Comments, most of the columns of each sample (comment) are set to zero conforming to a sparse dataset.

To compare the performance of models in different scenarios, we ran scenario pairwise comparison (theoretical—real/real—baseline) using Kappa and f-score measures. The comparison of theoretical and real scenarios is shown in Fig. 6 and allows measuring the impact of the quality and completeness of the data used to run e-SDRS on the effectiveness of the algorithms.

As shown in Fig. 6A, the use of sparse data (YouTube dataset) greatly affects the performance. However, when the dataset used for training is larger and its data are of higher quality (SpamAssassin dataset), the differences in the performance achieved in both scenarios are narrower (see Fig. 6B). The comparison between the real and baseline scenarios is shown in Fig. 7 and allows us to observe the impact in the performance of using e-SDRS.

As seen in Figs. 7A and 7B, the application of the e-SDRS algorithm is, in many cases, worthwhile (there is a small increase in performance). In some cases, there may be a small impact on some of the measurements performed. As an exception, we should mention the case of SVM used in conjunction with the YouTube Comments dataset. In this situation, we achieved relevant performance loss due to the sparsity of the instances included in it.

Using the optimal configuration settings for the e-SDRS algorithm ($MD = 3$, $RS = 80$), identified with the SpamAssassin and the YouTube corpora, we conducted an independent experiment using the Ling-Spam dataset. The train/test configuration is the same used in

**Table 3  Evaluation of e-SDRS performance in a real scenario.**

**(a) Youtube Comments Dataset**

| | MD2 | | | | MD3 | | | | MD4 | | | |
|---|---|---|---|---|---|---|---|---|---|---|---|---|
| | RS80 [3.08%] | RS85 [3.02%] | RS90 [3.02%] | RS95 [2.96%] | RS80 [6.76%] | RS85 [6.7%] | RS90 [6.64%] | RS95 [6.58%] | RS80 [11.05%] | RS85 [10.57%] | RS90 [10.27%] | RS95 [10.21%] |
| nb | 0.0000 | 0.0000 | 0.0000 | 0.0000 | +0.0227 | 0.0000 | 0.0000 | 0.0000 | +0.0971 | +0.1059 | +0.0786 | +0.0786 |
| ranger | −0.0275 | −0.0088 | +0.0178 | −0.0306 | +0.0046 | +0.0158 | −0.0416 | +0.0267 | +0.0411 | +0.0411 | +0.0197 | +0.0088 |
| rpart | −0.1122 | −0.0123 | −0.0911 | −0.0062 | +0.0127 | −0.0062 | −0.0229 | −0.1033 | +0.0296 | +0.0426 | +0.0089 | +0.0162 |
| svmPoly | +0.0873 | +0.0699 | −0.5674 | −0.3527 | −0.2548 | +0.0589 | +0.0675 | −0.5674 | −0.5674 | −0.4655 | +0.0515 | −0.5674 |
| treebag | −0.0213 | −0.0176 | +0.0135 | −0.0088 | +0.0100 | −0.0314 | +0.0024 | −0.0515 | +0.0503 | +0.0115 | +0.0201 | +0.0157 |

**(b) SpamAssassin Dataset**

| | MD2 | | | | MD3 | | | | MD4 | | | |
|---|---|---|---|---|---|---|---|---|---|---|---|---|
| | RS80 [6.42%] | RS85 [4.69%] | RS90 [3.95%] | RS95 [2.91%] | RS80 [6.87%] | RS85 [5.11%] | RS90 [4.28%] | RS95 [3.18%] | RS80 [7.14%] | RS85 [5.38%] | RS90 [4.48%] | RS95 [3.26%] |
| nb | +0.0861 | +0.0821 | +0.0676 | +0.0378 | +0.1447 | +0.0902 | +0.0711 | +0.0361 | +0.0888 | +0.0924 | +0.0757 | +0.0524 |
| ranger | +0.0147 | −0.0393 | −0.0304 | −0.0561 | +0.0116 | −0.0023 | +0.0144 | −0.0486 | −0.0320 | +0.0114 | −0.0177 | +0.0161 |
| rpart | +0.0203 | +0.0002 | −0.0188 | −0.0045 | +0.0359 | +0.0031 | +0.0161 | +0.0014 | +0.0252 | +0.0209 | +0.0147 | +0.0049 |
| svmPoly | −1.0833 | +0.0360 | +0.1278 | +0.1573 | +0.0977 | +0.0569 | −0.7733 | +0.0522 | −0.7733 | −0.3860 | +0.1417 | −0.1378 |
| treebag | −0.0203 | −0.0283 | −0.0376 | −0.0237 | −0.0001 | −0.0260 | −0.0406 | −0.0327 | −0.0045 | −0.0314 | −0.0406 | −0.0343 |

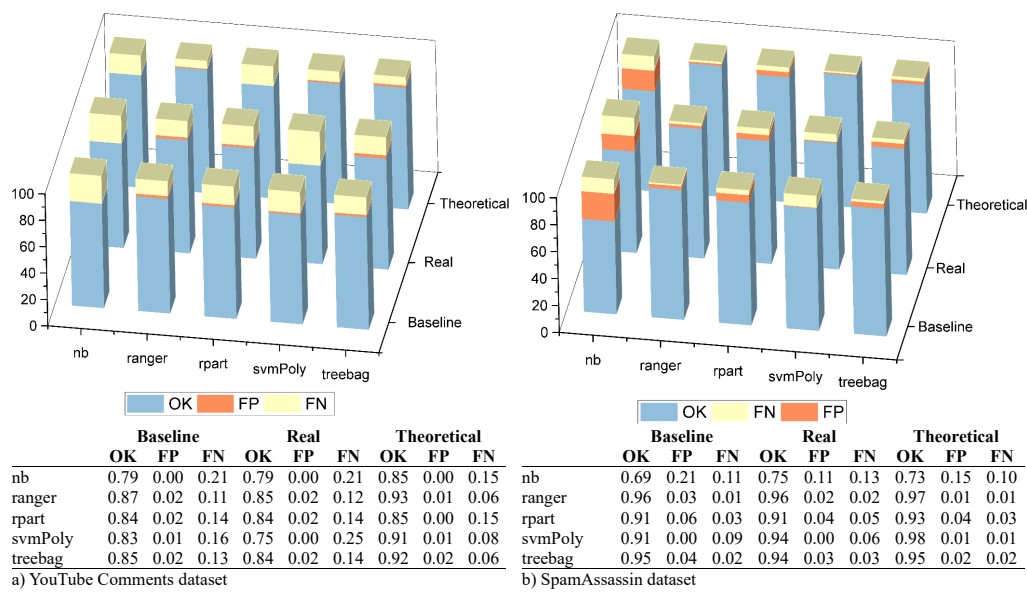

| | **Baseline** | | | **Real** | | | **Theoretical** | | |
|---|---|---|---|---|---|---|---|---|---|
| | OK | FP | FN | OK | FP | FN | OK | FP | FN |
| nb | 0.79 | 0.00 | 0.21 | 0.79 | 0.00 | 0.21 | 0.85 | 0.00 | 0.15 |
| ranger | 0.87 | 0.02 | 0.11 | 0.85 | 0.02 | 0.12 | 0.93 | 0.01 | 0.06 |
| rpart | 0.84 | 0.02 | 0.14 | 0.84 | 0.02 | 0.14 | 0.85 | 0.00 | 0.15 |
| svmPoly | 0.83 | 0.01 | 0.16 | 0.75 | 0.00 | 0.25 | 0.91 | 0.01 | 0.08 |
| treebag | 0.85 | 0.02 | 0.13 | 0.84 | 0.02 | 0.14 | 0.92 | 0.02 | 0.06 |

a) YouTube Comments dataset

| | **Baseline** | | | **Real** | | | **Theoretical** | | |
|---|---|---|---|---|---|---|---|---|---|
| | OK | FP | FN | OK | FP | FN | OK | FP | FN |
| nb | 0.69 | 0.21 | 0.11 | 0.75 | 0.11 | 0.13 | 0.73 | 0.15 | 0.10 |
| ranger | 0.96 | 0.03 | 0.01 | 0.96 | 0.02 | 0.02 | 0.97 | 0.01 | 0.01 |
| rpart | 0.91 | 0.06 | 0.03 | 0.91 | 0.04 | 0.05 | 0.93 | 0.04 | 0.03 |
| svmPoly | 0.91 | 0.00 | 0.09 | 0.94 | 0.00 | 0.06 | 0.98 | 0.01 | 0.01 |
| treebag | 0.95 | 0.04 | 0.02 | 0.94 | 0.03 | 0.03 | 0.95 | 0.02 | 0.02 |

b) SpamAssassin dataset

**Figure 5** Evaluation of hits, FP, and FN errors.

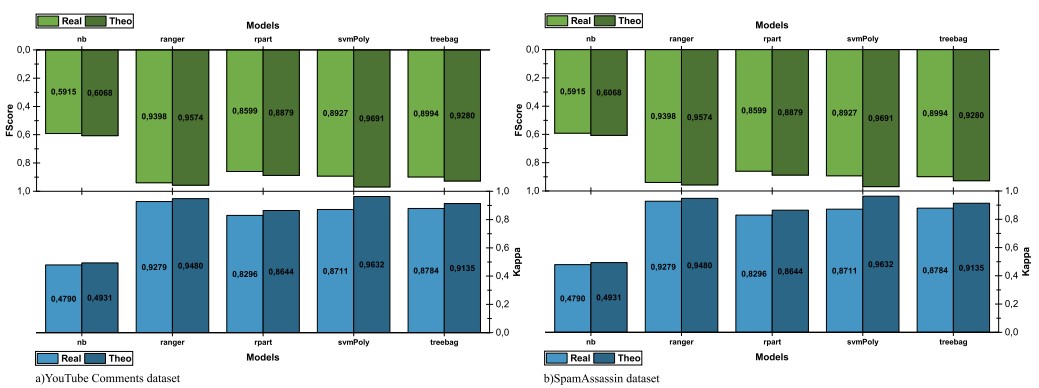

a)YouTube Comments dataset

b)SpamAssassin dataset

**Figure 6** Comparison of e-SDRS in theoretical and real scenarios.

the previous experiments (75%/25%). As the theoretical scenario has been designed for parameter optimization purposes, it has not been included in this experimental state. The evaluation metrics included the ratio of correctly classified messages (OK), error ratios (FP, FN), and f-score. Table 4 shows the performance of classifiers when using both the original dataset (baseline scenario) and the dataset reduced by using e-SDRS (real scenario).

Some results included in Table 4 (rpart and treebag algorithms in the baseline scenario) could not be computed due to the high dimensionality of the experiment. However, results support the observations of previous experiments. e-SDRS can be used to reduce dimensionality in datasets without losing performance or even slightly improving the accuracy of the algorithms. In addition, using the optimal parameters, e-SDRS was able to reduce the dimensionality from the original 39,174 synsets to 29,557 (24.5% of reduction).

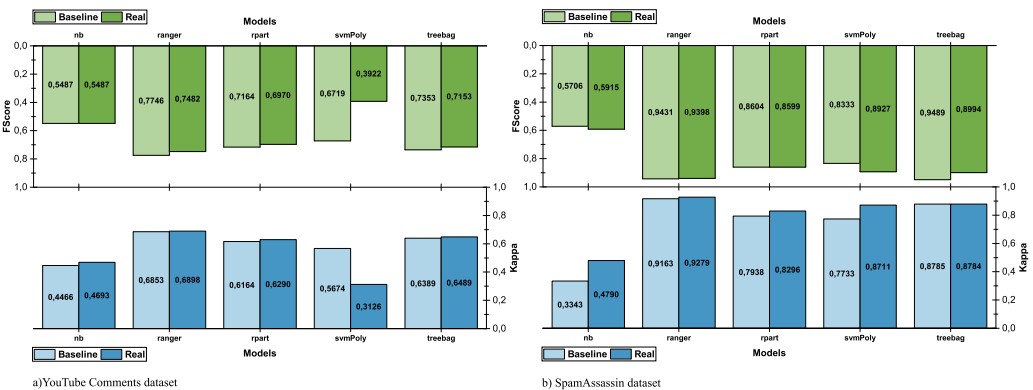

**Figure 7** Comparison of e-SDRS in baseline and real scenarios.

**Table 4** Evaluation of e-SDRS performance in a real scenario for LingSpam dataset.

| | Baseline | | | | Real | | | |
|---|---|---|---|---|---|---|---|---|
| | OK | FN | FP | f-score | OK | FN | FP | f-score |
| **nb** | 0.9549 | 0.0239 | 0.0211 | 0.8655 | 0.9986 | 0.0014 | 0 | 0.9958 |
| **ranger** | 0.9704 | 0.0183 | 0.0113 | 0.9106 | 1.0000 | 0 | 0 | 1.0000 |
| **rpart** | – | – | – | – | 0.9577 | 0.0211 | 0.0211 | 0.8750 |
| **svmPoly** | 0.9577 | 0.0211 | 0.0211 | 0.8750 | 0.9648 | 0.0211 | 0.0141 | 0.8936 |
| **treebag** | – | – | – | – | 0.9972 | 0 | 0.0028 | 0.9917 |

Finally, we compared the performance of e-SDRS with the original and the best SDRS variants (lossless and low-loss) (*Vélez de Mendizabal et al., 2023*). Similarly to the previous experimentation, the execution of SDRS was made using the 75% of the messages, while the remaining 25% ones were used for evaluation purposes. The comparison was made using accuracy and total cost ratio (TCR) (*Phuoc et al., 2009*) with different cost factors ($\lambda = 1$, $\lambda = 9$, $\lambda = 999$) measures. It is shown in Fig. 8. Due to the high computational requirements of SDRS variants (10 days), only the YouTube Comments dataset could be used in the evaluation. The analysis was carried out by comparing the best configuration achieved by both proposals.

As shown in Fig. 8, the performance achieved by lossless and low-loss SDRS variants are quite poor compared to our proposal. One of the most important benefits reported by the authors of SDRS is the ability to reduce FP errors. However, as seen in all the TCR measurements, the number of FP-type errors achieved by e-SDRS is considerably lower.

Finally, we also compared the computational requirements of both feature reduction schemes. SDRS variants were executed in a high-performance computer (4x4 Intel Xeon E7-8890 and 1 TB of RAM) while e-SDRS was executed in a lower-capacity one (Intel Core i7-6700 CPU 3.40 GHz × 8 and 64 GB of RAM). The average time required for executing e-SDRS with each configuration in YouTube Comments and SpamAssassin datasets were 9.62s and 2 h 52m, respectively (SDRS take 10 days when used to reduce
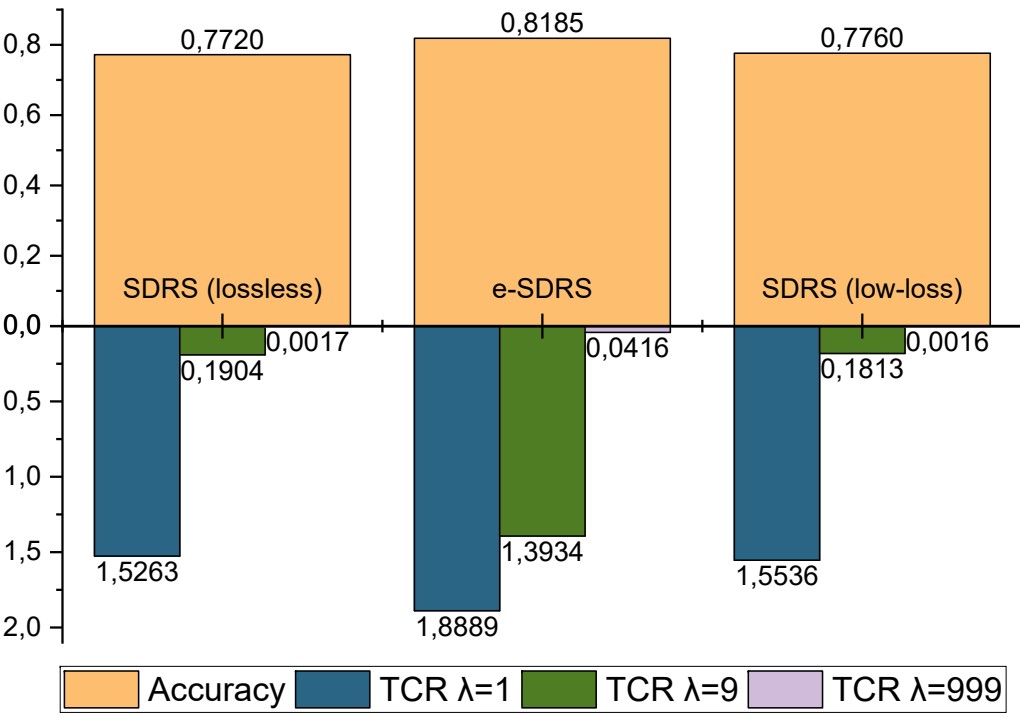

**Figure 8** Comparison between lossless and low-loss SDRS variants and e-SDRS performance.

the dimensionality of YouTube Comments dataset). Finally, e-SDRS executed the feature reduction of Ling-Spam corpus in 2 h and 44 min.

Considering the above results, we believe that the e-SDRS method is a reliable solution for rapidly reducing dimensionality. The next section shows the main conclusions and outlines the directions of future research.

## CONCLUSIONS AND FUTURE WORK

This work introduces a novel information fusion approach for the reduction of dimensionality in synset-based representations of data called e-SDRS. The proposal is based on combining some semantically related features (synsets). The identification features that can be combined are guided by hyponymy/hypernymy relations between synsets (achieved from an ontological dictionary, Babelnet) and the classes of the instances containing such features. In summary, two synsets could be combined when they are semantically close (there is a small path between them in Babelnet using taxonomic relations) and most documents containing any of them are included in the same category. In this study, we have analysed the performance of the method and compared it with two variants of a previous approach called SDRS, which is based on MOEA.

For the experiment, we selected three different datasets and analysed the performance of five well-known classifiers in three scenarios (theoretical, real and baseline). In the baseline scenario, e-SDRS was not applied. Moreover, the theoretical scenario is designed

to estimate the maximum capacity that the e-SDRS algorithm could have when the input dataset is particularly good. The real scenario tries to show the accuracy of the proposal in a normal situation. The selected corpora are SpamAssassin, YouTube Comments dataset and Ling-Spam. The second is a sparse dataset, a feature that severely affects the performance of e-SDRS and classification algorithms. Although e-SDRS showed adequate performance for all datasets, the experimentation allowed the differences between the maximum performance reachable by e-SDRS (assessed in theoretical scenario) and its performance in a real situation (real scenario) to be closer when using a better input dataset (SpamAssassin).

Although the application of e-SDRS requires the definition of some mathematical restrictions (condition for establishing whether two synsets are semantically related, criterion of whether two synsets appear mainly in documents of the same category), its application is computationally simple. As the combination of features in e-SDRS is guided by semantic information extracted from an ontological dictionary, some classifiers may perform slightly better. In the same vein as the SDRS approach, our proposal can be categorized as a lossless feature reduction schemes.

Although e-SDRS has provided improvements in lossless feature reduction schemes, some challenges in this area remain unaddressed. In particular, (*i*) the extension of the algorithm for addressing multiclass or multilabel classification problems and (*ii*) the exploration of non-taxonomic semantic relationships to improve dimensionality reduction, have not yet been contemplated. We believe that this is a promising direction in which we should move forward to find new mechanisms to reduce dimensionality and add useful semantic information in the classification process. Finally, there are some minor adaptations that would be interesting for e-SDRS, such as its adjustment to be used in multi-class and multi-label classification problems.

## ACKNOWLEDGEMENTS

SING group thanks CITI (Centro de Investigación, Transferencia e Innovación) from the University of Vigo for hosting its IT infrastructure.

### Funding

This work was endorsed by the project Semantic Knowledge Integration for Content-Based Spam Filtering (grant number TIN2017-84658-C2-1-R) from the Spanish Ministry of Economy, Industry and Competitiveness (SMEIC), State Research Agency (SRA) and the European Regional Development Fund (ERDF). In addition, this work was supported by the Conselleria de Cultura, Educación e Universidade (Xunta de Galicia) under the scope of the strategic funding of Competitive Reference Group (grant number ED431C 2022/03-GRC). The funders had no role in study design, data collection and analysis, decision to publish, or preparation of the manuscript.

## Grant Disclosures

The following grant information was disclosed by the authors:

Semantic Knowledge Integration for Content-Based Spam Filtering: TIN2017-84658-C2-1-R.

Spanish Ministry of Economy, Industry and Competitiveness (SMEIC).

State Research Agency (SRA).

European Regional Development Fund (ERDF).

Conselleria de Cultura, Educación e Universidade: ED431C 2022/03-GRC.

## Competing Interests

The authors declare there are no competing interests.

## Author Contributions

- María Novo-Lourés conceived and designed the experiments, performed the experiments, performed the computation work, authored or reviewed drafts of the article, and approved the final draft.
- Reyes Pavón analyzed the data, authored or reviewed drafts of the article, and approved the final draft.
- Rosalía Laza analyzed the data, authored or reviewed drafts of the article, and approved the final draft.
- José R Méndez conceived and designed the experiments, performed the experiments, performed the computation work, authored or reviewed drafts of the article, and approved the final draft.
- David Ruano-Ordás conceived and designed the experiments, analyzed the data, prepared figures and/or tables, authored or reviewed drafts of the article, and approved the final draft.

## Data Availability

The code implementing the experimental protocol used for the realisation of the article is available at Zenodo: Novo, M. (2024). An Enhanced Algorithm for Semantic-Based Feature Reduction in Spam Filtering Example Code. Zenodo. https://doi.org/10.5281/zenodo.10517957.

The datasets used to perform all the experiments are available at Zenodo: Novo, M. (2024). eSDRS datasets [Data set]. Zenodo. https://doi.org/10.5281/zenodo.12634056.

The source code of eSDRS technique used to perform the experiments (synset-based feature extraction stage) is available at GitHub and Zenodo:

https://github.com/sing-group/eSDRSexample/

mnloures. (2022). sing-group/eSDRSexample: Release 1.0 (1.0). Zenodo. https://doi.org/10.5281/zenodo.5949804.

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
