# Peer review of "An enhanced algorithm for semantic-based feature reduction in spam filtering"

_PeerJ Computer Science, doi:10.7717/peerj-cs.2206_

## Round 0.1 · original submission · Major Revisions

Please, read carefully the reviewers' comments and give a point to point response to address the concerns.

Reviewer 1 ·

Basic reporting

This manuscript proposes a new loss feature reduction scheme, which can achieve higher accuracy in FP error and use fewer computing resources. In addition, the manuscript also proposes a new effective semantic-based dimensionality reduction algorithm for binary classification problems (e-SDRS). Experimental results have shown that by using this dimensionality reduction algorithm, classifiers can achieve better accuracy and have lower computational requirements.

Experimental design

Unfortunately, there are still some issues with this manuscript.

1. The comparative dimension of the experiment is insufficient. The author needs to compare more computing resources to prove the progressiveness of the algorithm.

Validity of the findings

2. There are many syntax errors, such as (i) The study includes experiments using two datasets (small and medium sizes) and a detailed comparison with two previous optimization based approaches ->approaches (ii) These techniques have been successfully applied to solve a wide variety of problems with specific constraints including spam filtering, language detection, or->and news categorization (Kowsari et al., 2019).

Cite this review as

Reviewer 2 ·

Basic reporting

All comments are written in the "Additional comments" section.

Experimental design

All comments are written in the "Additional comments" section.

Validity of the findings

All comments are written in the "Additional comments" section.

Additional comments

In this study, spam mail detection, one of the text classification problems, is discussed. One of the difficult tasks in text classification is the dimension reduction/feature selection/feature reduction task. The authors propose a semantic features-based approach for dimensionality reduction. The proposed approach was developed for two-class problems. My comments about the work are as follows:
- The literature research section in Chapter 2 should be expanded by focusing on current studies.
- No details are given regarding the experimental adjustments section. More details are needed. For example, what was done in the pre-processing steps? Which tools were used for the pre-processing step? Was a term weighting strategy used? If so, what term weighting method was used? Which of the document vector representations was used? Details like this should be given.
- Baseline, Real and Theoretical scenarios described in the article are not fully understood. What these are can be explained more clearly.
- What is the total number of words in the data sets used? How do classification algorithms perform when classification is made with all word numbers? On the other hand, the number of words is reduced by the proposed technique. What is the result achieved with few (reduced) words? Providing detailed results like this will reveal the effect of the proposed method more.
- The proposed approach can be compared with some of the filter and wrapper feature selection techniques used in text classification problems. They are not feature selection/dimensional reduction approaches with the same characteristic feature reduction methods. But giving a result in terms of its effects on classification will be very important for researchers.
- I recommend comparing the results obtained with the results in the literature (specific to the same data sets).

Cite this review as

---

## Round 0.2 · accepted · Accept

I have evaluated the comments of the reviewers, and your responses. I have decided to accept the article. Congratulations on the acceptance of your manuscript!